# The influence of wind veer on fatigue loading for large floating wind turbines with flexible drivetrains

Veronica Liverud Krathe<sup>1</sup>, Jason Jonkman<sup>2</sup>, and Erin E. Bachynski-Polić<sup>1</sup>

<sup>1</sup>Department of Marine Technology, Norwegian University of Science and Technology (NTNU), Trondheim, Norway <sup>2</sup>National Renewable Energy Laboratory (NREL), Golden, Colorado, United States **Correspondence:** Veronica Liverud Krathe (veronica.l.krathe@ntnu.no)

**Abstract.** To reduce cost, offshore wind turbines are expected to be designed with significantly increased rotor diameters. Larger turbines become more flexible and span a larger portion of the atmospheric boundary layer. With these changes, the validity of traditional modeling assumptions should be investigated. This work challenges two common assumptions: 1) that the drivetrain can be considered rigid (except in torsion) and does not couple with the rotor and tower and 2) that wind directional

5 change with height (veer) does not greatly influence the fatigue damage in the tower, blades and drivetrain.

Two large floating wind turbines are considered: the International Energy Agency (IEA) 15 MW (Gaertner et al., 2020) reference turbine with the University of Maine VolturnUS-S platform (Allen et al., 2020) and the IEA Wind 22 MW reference turbine (Zahle et al., 2024a). Both are direct-drive generator turbines supported by semisubmersible platforms. Aero-hydroservo-elastic simulations are performed using OpenFAST, with drivetrain bending flexibility and main bearing response imple-

- 10 mented in the coupled analysis. The turbines are subjected to a set of load cases at below-, near- and above-rated mean wind speeds, assembled based on NORA3 hourly wind and wave hindcast data (Haakenstad et al., 2021) for Utsira Nord off the coast of Norway (Cheynet et al., 2023, 2024). Within each load case, conditions with and without veer are simulated to evaluate the influence of veer on damage equivalent loads (DELs) of the turbine tower, blades and main bearings. Further, these load cases are applied to evaluate the influence of drivetrain flexibility on global turbine response.
- The results indicate that, depending on the veer gradient, operating regime and turbine size, veer can be very important for DELs of the tower top and blade root and for the fluctuations of the main bearing radial loads. Moreover, drivetrain flexibility is found to influence global DELs, especially for the largest turbine. Comparing flexible and rigid drivetrains, the tower-top fore-aft and torsional damage equivalent moments of the 22 MW turbine are reduced by more than 20 % at near rated wind speeds. For the same turbine in below-rated wind speeds, the blade root flapwise DELs are reduced by up to 6 %.

# 20 1 Introduction

Offshore wind turbines are expected to reach rated capacities of 15 MW and beyond (IRENA, 2019). With increasing turbine size, the applicability of current modeling approaches becomes more uncertain. Mechanisms that have been neglected in the design of smaller turbines may play an important role for larger wind turbines. Increased structural flexibility is one feature that introduces new demands on state-of-the-art analysis tools. Another challenge emerges from the rotor extending deeper into, or

even beyond, the atmospheric boundary layer (ABL), so that previous assumptions related to the wind profile and wind field can be questioned. In this work, both of these topics are investigated.

Based on the assumption of a relatively rigid drivetrain, main bearing response has traditionally been obtained by two separate analyses. First, a global analysis, with the drivetrain represented by a torsional spring and damper, outputs shaft loads. These shaft loads are then combined with either analytical calculations or a detailed local model of the drivetrain to obtain main

- bearing loads. Analytical calculations are fast but complex to derive for bearings that have off-diagonal or moment-carrying stiffness terms. Local models are accurate but computationally expensive and time-consuming to develop. Additionally, the decoupling of drivetrain flexibility from global turbine response becomes questionable as turbines become larger and more flexible (Torsvik, 2020). For instance, Wang et al. (2021) found non-torsional modes of the drivetrain with lower natural frequencies than the torsional mode when considering a fully coupled drivetrain-turbine model.
- In design, the wind profile has typically been assumed as sheared (mean speed varying with height) but without variation 35 of mean wind direction with height. A common definition of wind "veer" is the clockwise rotation of the mean wind direction 36 with height, while "backing" refers to winds rotating counterclockwise with height (Lundquist, 2022). Veer is a phenomenon 37 often associated with stable boundary layers (Stull, 1988). Shu et al. (2020) found the largest veer angles for neutrally strat-38 ified boundary layers using lidar measurements off the coast of Hong Kong. Wind veer often stems from the influence of
- frictional forces on the force balance. The frictional forces are present in the ABL, diminish with height, and are not present in the troposphere. Veer can also be a result of inertial oscillation in the ABL or horizontal temperature gradients (Lundquist, 2022). Offshore measurements near Martha's Vineyard collected over 13 months showed that strong wind veer often occurred, especially in the spring and summer and with low wind speeds (average values of up to 0.1° m<sup>-1</sup> and extreme values above  $0.3^{\circ}$  m<sup>-1</sup>) (Bodini et al., 2019, 2020). Veer and backing occurred 70 % and 30 % of the time, respectively. A similar ratio of
- backing to veer was found by Marini et al. (2025) based on 12 months of lidar measurements in Belgian waters (excluding June and August) and with occurrences of high veer gradients above  $0.3^{\circ}$  m<sup>-1</sup>. Shu et al. (2020) also found decreasing veer gradient with increasing wind speed.

Two studies investigated the sensitivity of a 5 MW wind turbine to various modeling parameters; Robertson et al. (2019) considered an onshore variant while Wiley et al. (2023) considered the same turbine atop a semisubmersible. Robertson et al.

- (2019) observed that turbine fatigue loads (tower, blade-root and main shaft bending moments) were less influenced by veer than by turbulence and shear. Similarly, Wiley et al. (2023) found low sensitivity of fatigue-proxy loads to veer compared to wind velocity standard deviation, turbulence coherence and the wave conditions. On the other hand, Hart et al. (2022) found that veer significantly influenced the main bearing radial load fluctuations, more so than the other deterministic effects they evaluated (shear, yaw offset, mean wind speed), for a 5 MW onshore turbine. They used a simplified load response model and
- deterministic wind fields. Larger veer led to larger fluctuations, while negative veer (backing) of the same magnitude led to a smaller increase in radial load fluctuations. Applying similarity scaling to 7.5 MW and 10 MW turbines, veer was seen to scale more than cubically with rotor radius. To the authors' knowledge, there are no studies on the influence of veer on turbine loads for turbines larger than 10 MW, and the IEC 61400-1 wind turbine design standard (IEC, 2019) does not mention veer, even

75

though it is an inherent characteristic of the wind field. A rotor spanning 200 to 300 m can potentially sample a large variation of wind directions, and it is important to understand the effects of veer on turbine loads.

This paper addresses the influence of increased drivetrain flexibility and wind veer on damage-equivalent loads (DELs) in the tower, blade root and main bearings for two floating turbines. A coupled drivetrain-turbine model was built in OpenFAST. This modeling approach was previously verified against a multibody model (Krathe et al., 2025a) and applied in investigations of wake effects on main bearings (Krathe et al., 2025b). The model includes a flexible bedplate and shaft, both represented by

- linear beams, and flexible main bearings are included by means of constant stiffness matrices. The model enables investigation of the influence of drivetrain flexibility on the turbine response. It also allows for inclusion of main bearing off-diagonal stiffness terms and direct extraction of main bearing response from global analyses. With regards to veer, reanalysis data from NORA3 (Haakenstad et al., 2021; Cheynet et al., 2024, 2023) were applied to estimate appropriate veer gradients and accompanying turbulence intensities, shear profiles and wave conditions for four different mean wind speeds for the Utsira
- Nord site, designated for floating wind farms.

Section 2.1 describes the reference wind turbines and the flexible drivetrain model. Section 2.2 presents the load cases and the derivation of the environmental conditions. Section 2.3 explains the calculation of DELs. Section 3 presents and discusses the results, including modal analysis of the turbines with and without flexible drivetrains (Sect. 3.1), influence of drivetrain flexibility on turbine response (Sect. 3.2), and the DELs of the tower, blades and main bearings subjected to various wind fields (Sect. 3.3). Conclusions are summarized in Section 4.

#### 2 Methodology

#### 2.1 Wind turbine model

# 2.1.1 Base cases and numerical setup

Two floating reference turbines, the International Energy Agency (IEA) 15 MW (Gaertner et al., 2020; Allen et al., 2020) and
the IEA 22 MW turbine (Zahle et al., 2024a), were considered in this work. Both have direct-drive generators and are supported by semisubmersible floating platforms with one centered column, three radially spaced columns and three catenary mooring lines. The main properties of the turbines are listed in Table 1, and the drivetrain and floater of the IEA 15 MW turbine are shown in Fig. 1.

The turbines were modeled in the aero-hydro-servo-elastic code OpenFAST (Jonkman et al., 2024). A ROSCO controller (Abbas et al., 2024) was applied for each turbine. For such large rotors, it is generally recommended to account for large deflections in OpenFAST by modeling the blades using the geometrically exact beam theory in the module BeamDyn. Unfortunately, it was not computationally feasible to combine BeamDyn and the drivetrain model described in Sect. 2.1.2 for the floating turbines. Instead, ElastoDyn was applied, using a modal representation of the blades. Tower influence on wind speed was included using potential flow theory with the Bak correction (Bak et al., 2001).

#### Table 1. Main parameters of the floating turbines

| Turbine   | Hub height<br>[m] | Rotor diame-<br>ter [m] | Rated wind speed $[m s^{-1}]$ | Rated power<br>[MW] | Rated rotor speed [rpm] | Shaft<br>tilt angle [°] | Water<br>depth [m] |
|-----------|-------------------|-------------------------|-------------------------------|---------------------|-------------------------|-------------------------|--------------------|
| IEA 15 MW | 150               | 240                     | 10.59                         | 15                  | 7.56                    | 6.0                     | 200                |
| IEA 22 MW | 170               | 284                     | 11                            | 22                  | 7.06                    | 6.0                     | 200                |

Figure 1. The IEA 15 MW reference turbine (Krathe et al., 2025b) and its direct-drive generator (Gaertner et al., 2020). The IEA 22 MW turbine has a similar design.

#### 90 2.1.2 Drivetrain model

In this work, the two reference OpenFAST models (Barter et al., 2024; Zahle et al., 2024b) were modified to include a flexible drivetrain, modeled together with the tower in the OpenFAST linear finite element module SubDyn (NREL, 2024). This methodology was previously verified against a coupled multibody model and applied for a 10 MW geared drivetrain (Krathe et al., 2025a) and for the IEA 15 MW direct-drive turbine (Krathe et al., 2025b). In short, a flexible drivetrain was included in

- the coupled, global analyses. Flexible beams were applied to represent the bedplate and shaft, and constant springs represented the main bearings. Additional drivetrain mass and inertia were modeled using point masses. More details of this modeling approach can be found in the two aforementioned papers and on GitHub (Krathe, 2024). For comparison, OpenFAST models with rigid drivetrains and flexible towers were also built in SubDyn. Figure 2 presents the two versions of the global model with flexible and rigid drivetrains.
- For the flexible drivetrain, the bedplate and shaft geometry and material properties were obtained directly from the WISDEM framework for each turbine (Barter et al., 2024; Zahle et al., 2024b). The bearings are taken as two spherical roller bearings

105