# Peer review of "The influence of wind veer on fatigue loading for large floating wind turbines with flexible drivetrains"

_Wind Energy Science, 2025_

## Referee Comment (RC1)

The reviewer strongly believes that the paper presents insights into fatigue loading for large floating wind turbines with flexible drivetrains, but not a lot of focus was given to the veer aspect of the study. The results appear original and are well written.

Page 1: Abstract: This is a nontechnical review comment – In academic writing, an abstract does not include intext references (e.g Gaertner et al., 2020, Allen et al., 2020, etc). Rewrite to suit.

Page 1: Introduction: Line 24: It would be beneficial to state or quantify the layer in which you are determining the ABL (in the form of an elevation, say, 100-300m or whether it is stable or unstable, or NBL)

Page 1-3: Introduction: The introduction so far is well structured. The roadmap is clear. However, where you have descriptions like "becomes more uncertain" or "questionable", it would be best to quantify this uncertainty where possible.

Page 1-3: Introduction: The introduction so far is well structured. So, the roadmap is clear. However, where you have descriptions like "becomes more uncertain" or "questionable", it would be best to quantify this uncertainty where possible.

Page 3-10: Introduction: Make a bold statement about why semisubmersible only was chosen. It will prevent questions about other floating systems

Page 5: Methodology: Line 106: The reviewer is interested in how the kxx, kyy, kxy, and kzz is mapped into SubDyn spring elements. Specifically, how you rotate the off diagonal from the local coordinate frame of the bearing into SubDyn's element axes. This might interest the general reader too, so consider including it in the paper.

Page 3-10: Methodology: Line 115: Not critical, but is there a reason why the time steps differ between 15MW and 22MW models? Was dt chosen to satisfy stability and accuracy in the Craig–Bampton RO model?

Page 7: Methodology: Line 147: Are semisubmersibles not sensitive to directional spreading, since you use a Pierson-Moskowitz spectrum for wave generation. Did you model the wave as unidirectional.

Page 9: Methodology: Table 6:  Is there a reason why your grid width (264/316 m) only barely exceeds the rotor diameter?

Page 14-19: Result and discussion: You present DELs for SV, ST, SVT, and SBT, but without discussing the linearity or complexities of the wind veer cases, readers cannot understand the incremental effect of veer itself. There should be a baseline case (i.e., without veer) for

comparison. I believe this is what the study meant by analyzing the influence of veer. This is the comparison the reviewer expects. Otherwise, the topic could simply be "Fatigue loading for large floating wind turbines with flexible drivetrains."

Page 18: Result and discussion: It would be better to keep the y-axis uniform in Figures 12, 13, and 14, as it appears the magnitudes are the same when it is not.

Page 20: Conclusion: Again, to be clear, you rightly discussed the effect of veer, perhaps using your prior knowledge of the topic, but did not provide details on how veer drives DEL increases. The methods and results lack a clear, standalone "veer-only" case and do not show how veer was implemented or isolated in your analysis.

---

## Referee Comment (RC2)

Review of wes-2025-92 "The influence of wind veer on fatigue loading for large floating wind turbines with flexible drivetrains"

Dr Edward Hart

**Dear authors,**

This paper seeks to investigate the influences of drivetrain flexibility and wind veer on wind turbine damage equivalent loads, considered here in the context of blades, tower and main bearings. It undertakes this analysis in the context of large offshore floating wind turbines (22 MW and 15 MW). Results indicate that wind veer can increase DELs by as much as 30-70% for some components under some sets of wind conditions, and that accounting for flexibility of the drivetrain can result in DEL reductions on the order of 6-22% under some conditions.

While this manuscript certainly contains interesting results, I believe extensive revisions are required before it can be deemed ready for publication. My high-level concerns are as follows:

1) The paper lacks a clearly defined and specific set of research goals or questions, along with a clear narrative around why we're interested in this particular set of components, phenomena and metrics. A variety of analyses are undertaken, but the current exposition only links them fairly tenuously. By linking to prior work, motivating what is undertaken, and more narrowly defining objectives the paper can become more focussed, and its contribution made clearer (specific instances of this are highlighted throughout my "specific comments" below).

2) Related to the above, various results return only minor sensitivity to the effects under investigation, whereas in some cases the effects are significant. I'd suggest focussing on the latter, and possibly digging deeper in those cases to extract more insight and value.

3) The paper lacks a thorough treatment of relevant background research and associated contexts. The result of this is a large prior-knowledge burden placed on the reader, and the risk of some readers mis-interpreting findings, results and implication.

4) There are various point in the paper where assumptions and modelling decisions need clarification or better explanation/justification.

5) Finally, I feel some results figures need enhancing (e.g. use all seeds rather than just 1), some seem not to add much value and so might be safely removed (spectra), and some additional plots need adding (mean load plots for MB results).

To be clear, I am confident that good work has been undertaken which will be of value to the wind community. But, I believe the current manuscript falls short of what is required from a research publication.

**Specific comments**

Abstract – some bold claims are made here which I'm not sure the results quite live up to. Please consider a slightly more measured summary of paper outcomes in the abstract

General comment – the title is maybe misleading, as it highlights veer as the headline effect under investigation. Please consider a more representative title for the work which is presented.

General comment – there seems to be a general conflation between classic and rollingcontact fatigue in the paper (or at least the distinction is never made). These are not the same mechanism!

General comment – some amount of drivetrain flexibility is modelled, and its effects on dynamics and loads evaluated. But, it is also important to appreciate that the bearing housing is not modelled. If this component deforms during operation then additional interactions may occur which shorten bearing life. So, when describing what new effects are modelled, I'd encourage you to also highlight those effects which remain un-modelled.

Line 21 – "With increasing turbine size, the applicability of current modelling approaches becomes more uncertain. Mechanisms that have been neglected in the design of smaller turbines may play an important role for larger wind turbines. Increased structural flexibility is one feature that introduces new demands on state-of-the-art analysis tools." Please add some relevant references to back up these claims. This is indeed a much considered topic and so there will be useful citations to include here.

Line 24 – "Another challenge emerges from the rotor extending deeper into, or even beyond, the atmospheric boundary layer (ABL), so that previous assumptions related to the wind profile and wind field can be questioned." Again, please use some of the many relevant references to back up your claims.

Line 25 – "In this work, both of these topics are investigated." I don't think it's valid to claim this analysis specifically considers extension beyond the ABL, and since kinematic (Kaimal spectrum) wind fields are utilised, I'd also advise being careful when making claims about how representative these simulations are of the ABL.

Line 27 – "Based on the assumption of a relatively rigid drivetrain, main bearing response has traditionally been obtained by two separate analyses. First, a global analysis, with the drivetrain represented by a torsional spring and damper, outputs shaft loads. These shaft loads are then combined with either analytical calculations or a detailed local model of the drivetrain to obtain main bearing loads. Analytical calculations are fast but complex to derive for bearings that have off-diagonal or moment-carrying stiffness terms. Local models are accurate but computationally expensive and time-consuming to develop." The text guoted here is, as far as I can tell, the sum-total of background discussion provided on main bearing modelling work. No references are given to back up any of the claims made and, therefore, significant volumes of prior work have been completely ignored. This is highly problematic and leaves the reader without any appreciation for the extant work in this field. An important implication of this is that the reader now also lacks any appreciation for the nuances which remain present in main bearing research, i.e. we don't yet have a clear picture of the dominant failure mechanisms and so much ongoing modelling and data analysis work is specifically in aid of identifying candidate mechanisms (internal to the bearing and/or as a result of inflow conditions and load characteristics). Further work in this domain (e.g. investigating possible veer influences) is only valuable if carefully couched within the full context of main bearing R&D efforts and current understanding. Additionally, in the current exposition, it's not clear that the "global analysis" you describe is the aeroelastic-code (FAST, Bladed, Hawc2 etc). Again, clearer writing and proper referencing is needed. For example, you describe the global analysis as representing the drivetrain by a torsional spring and damper; is this true of all/most aeroelastic codes, or is that just FAST?

Many relevant main bearing papers can be found in WES, Wiley Wind Energy, and Energies journals. The following seem particularly relevant (to be transparent, I am co-author on two of these. Entirely up to you which – if any – you include):

- Electric Power Research Institute (EPRI). (2024) Wind turbine main bearing reliability analysis, operations, and maintenance considerations. Technical report 3002029874. https://www.epri.com/research/programs/113055/results/3002029874
- Hart, E., et al. (2023). Main bearing replacement and damage field data study on 15 gigawatts of wind energy capacity. Technical Report NREL/TP-5000-86228. https://docs.nrel.gov/docs/fy23osti/86228.pdf
- Kenworthy, J., et al. (2024). Wind turbine main bearing rating lives as determined by IEC 61400-1 and ISO 281: a critical review and exploratory case study. *Wind Energy*, 27(2), 179-197. (also see Zaretsky 2016 referenced therein)
- Sadeghi, F., et al. (2009). A review of rolling contact fatigue, J. Tribol., 131, 041403.
- Tallian, T. E. (1992) Simplified Contact Fatigue Life Prediction Model Part I: Review of Published Models, J. Tribol., 114, 207–213.
- Feiyu Lu, et al. (2024). A quantitative approach for evaluating fatigue damage under wake effects and yaw control for offshore wind turbines, Sustainable Energy Technologies and Assessments, Volume 66, 103824.

Beyond the lack of exposition concerning prior main bearing literature and modelling work, there is also a lack of similar discussion concerning all components and analysis metrics used. To put this another way, you don't motivate why the study is focussed on blades, towers and the main bearing (when there is extensive literature motivating each as important), and there is no discussion of the nuances of the DEL metric (and for the MB the DEL is based on L10 values, which have another raft of caveats associated with them) and how informative/restrictive it is. These last points all represent significant limitations in our ability to translate simulated loading into expected real world lifetimes and outcomes for components. That doesn't stop the study being valuable, but you have to clearly communicate that these limitations are present. I'd also add that these discussion don't need to be lengthy, and can mostly highlight key points and then refer to relevant literature, but they must be present.

Line 39 – "Wind veer often stems from the influence of frictional forces on the force balance"... what forces balance?

"Veer can also be a result of inertial oscillation in the ABL or horizontal temperature gradients"

The above two quotes aren't very intelligible to someone not already familiar with these concepts, can you please develop these points so they may be more readily understood.

Line 61 – "This paper addresses the influence of increased drivetrain flexibility and wind veer on damage-equivalent loads (DELs) in the tower, blade root and main bearings for two floating turbines" There is no motivation given for why we're interested in drivetrain flexibility and veer in particular? Why these two phenomena in particular?

Building on the above point, I'd add that the introduction is not very well crafted as it currently appears. More specifically, there is a lack of structured narrative to help the reader follow why we're interested in these specific combinations of components and effects, and where we are relative to prior work. One structuring issues example is that the introduction covers drivetrain flexibility and upscaling, then main bearing modelling, then wind veer. We're therefore jumping between big picture and small details, and without ever linking drivetrain flexibility with veer. Please revise the structure and content of the introduction in order to clearly and explicitly arrive at well-motivated research questions that naturally lead into the study which has been undertaken.

Line 62 – "A coupled drivetrain-turbine model was built in OpenFAST... etc" This is Methodology or Background content, and should not appear in the Introduction. Indeed, considering this alongside previous comments, perhaps what this paper lacks most is a proper Background section in which past work and relevant literature can be properly presented and discussed.

Line 85 – "For such large rotors, it is generally recommended to account for large deflections in OpenFAST by modeling the blades using the geometrically exact beam theory in the module BeamDyn. Unfortunately, it was not computationally feasible to combine BeamDyn and the drivetrain model described in Sect. 2.1.2 for the floating turbines". Isn't this a major issue, given the current study is specifically looking to understand loading on large flexible turbines? What if the errors induced by poor blade modelling are swamping any relevant effects from the flexible drivetrain or veer presence? It isn't enough to simply state that it wasn't possible, you need to seek to verify what the impact of this modelling decision will likely be on results. Please consider how this can be done. My suggestion would be to run simulations of both turbine models with the standard/simple drivetrain representation using both BeamDyn and Elastodyn (modal repr.), and then compare blade, tower and main bearing loads coming from each. If those differences are "small", then it provides some confidence that Elastodyn could be sufficient for the elastic drivetrain case. More generally, this simplification and its impact on results needs more thorough treatment in the paper.

Line 91 – "modified to include a flexible drivetrain, modeled together with the tower" This reads a little like the dynamic multibody interactions would only be between flexible drivetrain and tower. Is there also interaction between the flexible drivetrain and the rotor?

Line 102 - "and a downwind locating SRB" A locating SRB is necessarily double-rowed

(e.g. file:///C:/Users/edwar/Downloads/tpi\_251\_de\_en.pdf). This, and any general discussion of bearings/bearing-types is important contextual information that is currently missing from the paper. Also, why were SRBs chosen, and how were they sized etc... (maybe those details are in a previous paper, but they'd still be relevant here).

Line 103 – "The diagonals of the main bearing stiffness matrices..." Stiffness matrices have not been introduced, there is therefore a high burden placed on the reader to already know about these concepts.

Line 103 – "The diagonals of the main bearing stiffness matrices were estimated using Schaeffler's tool (Schaeffler, 2025) considering the force and deflection of the bearings for a number of different combinations of mean loads" Which bearings? If you have specific bearings being modelled, don't you also have the dynamic capacity and other information which would allow you to calculate L10 lives directly, rather than relying on relative damage? If you can then I believe you should, as then you can also account for the site wind speed distribution and combine all load conditions into a single resultant L10 life. Additionally, that would remove ambiguity between which bearing is expected to fail first (if it fails from rolling contact fatigue).

Table 2 – you utilise stiffness cross-terms, but it's not clear from the explication provided how they are utilised in the flexible drivetrain implementation. Please provide some context.

Line 109 – "Note that the main bearing stiffness estimates differ from those presented by Krathe et al. (2025b) due to updates in estimation methodology". What updates? Are the new values better?

Line 110 – "Note also that there are large uncertainties in the main bearing stiffness estimates". More fundamentally, bearing stiffness is a function of load. This follows from understanding that under increased load, more rollers are reacting the applied load. This is another point where the bearing context is absent, but would help elucidate the point being discussed. Was the validity of using a single linear spring to model the bearing under different load levels demonstrated in a previous paper? My concern it that the linear assumption could lead to large errors in some load cases. This is why in the past we've coupled our linearised spring model to a bearing contact model which estimates local stiffness as the load changes. For these same reasons, I'm not sure how informative the bearing stiffness sensitivity analysis is here, given the bearing stiffness will in fact change continuously throughout operation. Interpreting that sensitivity analysis should be done with these things in mind.

Line 116 – "glue-code (OpenFAST) time step of 0.0015 s and 0.0025 s, respectively." Why these values? I assume for convergence reasons. Did you arrive at these (if so how?) or are they standard values?

Line 134 – If you're interested in veer effects, it could obfuscate the results to also change the power law shear exponent. As it turns out the values across load cases appear to be almost identical, so I don't think this is an issue. But if they were different it would make it difficult to discern what was driving any changes which were observed.

Equation 3 – similar to the above comment, if you're interested in veer, don't let TI vary between each load case! (Again, they don't vary much so probably not a huge issue).

Section 2.2.3 – shouldn't this be within the previous subsection on Environmental Load Cases?

Section 2.3.1 – Please highlight here the fact that DELs allow for comparison between how damaging two sets of conditions are, and that they don't translate to a life estimate for that component. Also important to highlight that the concept for Loads is borrowed from the concept of Damage Equivalent Stress. Importantly, there is no good justification for reformulating the latter as the former, and that this is only truly valid if stress is proportional to the applied load. For turbine blades and tower this is highly likely to be true most of the time. DELs should therefore be highlighted as an indicative tool at best, and one for which we don't currently have a good alternative in general.

Line 171 – "Neq is the equivalent number of load cycles until failure" Under the calculate DEL

Line 173 – "The Wöhler exponent is taken as 10 for the blades and 4 for the tow" reference?

Line 181 – "These factors were estimated based on product catalogs" Earlier it seemed like you had known bearings from which you were modelling stiffness. Do you not in fact have that? What is it specifically you are getting from catalogues, I'm guessing the contact angle, or fatigue load limit (e)? Just to be clear, if you look in ISO 281 Table 8 it gives you a formula for e as a function of contact angle.

Table 7 – For the non-locating bearing Fa = 0 and (assuming contact angle is 0) e=0, so the relevant row should read 1, 0, NA, NA, 0.

Section 2.3.2 – this treatment of MB DELs lacks any discussion of the L10 life, ISO 281 formulations, or any of the surrounding context, caveats and uncertainties tied in here. Whether or not rolling contact fatigue (surface or subsurface initiated) is a dominant failure mechanism for main bearing in wind turbines remains unknown, beyond that it also remains

unclear to what extent ISO lifing methods provide accurate rolling contact fatigue lives given the size and complex nonsteady load conditions of these bearing (especially in large wind turbines). Beyond that, rolling contact fatigue for bearings is expressed as an L10 life, i.e. the time after which 10% of a population are expected to have failed. Another critical factor is the fat that, due to the nature of rolling contact fatigue and the ISO 281 method, main bearing L10 values are driven by the mean MB load rather than load fluctuations (discussed in Kenworthy 2024, listed previously). These are all important and occasionally subtle points which should accompany any application of these methods, to avoid misleading people to believe these metrics can be used to confidently assess main bearing longevity. DEL formulation does allow one to dispense with the bearing dynamic capacity (Cd), but again this should be highlighted to the reader via a proper discussion. Finally, equation 7 is simply presented without any derivation or explanation. Finally if t\_i is simply the timestep, then no subscript "i" is necessary.

Section 3.1 – is this a results section? Surely it's more appropriate within the methodology or an appendix?

Line 190 – "Aerodynamics were omitted and mooring lines were replaced by linear springs to facilitate detection of modes. Rigid-body mode natural frequencies were found through decay tests." These statements are made without context, justification or supporting references (if applicable). Again, a high burden is being unfairly placed on the reader here.

Figure 4 – Why are we only looking at results for a single seed? Pleased instead develop plots which summarise information across seeds. For example you could easily plot the mean, std, max and min across each set of 6 seeds at each wind speed. This would provide a more robust picture of impacts here.

Line 218 – "It is difficult to compare main bearing loads between the rigid and flexible..." It feels like we're back into methodology here, why isn't this discussion had in the methodology section?

Line 220 – "As a proxy for main bearing loads, the standard deviations of the shaft axial load and pitch and bending moments are compared among the two models". As mentioned previously, MB rating life (L10) and hence DEL\_MB is principally driven by the mean MB load and not by standard deviations. This is counterintuitive, and why a detailed treatment of bearing and rolling contact fatigue complexities are essential. Please undertake the same analysis (summarising across all seeds) for the mean loads as well as std.

Figures 4 and 5 – When revising these figures, please use the same fixed axes on all plots in order to make them directly comparable to each other without careful study of axis scaling.

Section 3.2.2 – This section is more of a verification that the uncertainty around bearing stiffness is likely not driving large errors in the results. As such I'd suggest moving it to an appendix, as it distracts from the main focusses of the paper.

Figure 7 – as mentioned above, it is mean loads that are mostly going to drive your MB DEL. Please also develop figures showing changes in mean loading as you change bearing stiffness. This should include Fa, Fr and P values. As above, it seems this has only been plotted for a single wind seed at each point, please therefore also represent results for all seeds here as well.

Line 254 – what is the logic behind including a case for shear and veer without turbulence? I can't see why that adds anything?

Section 3.3.4 – I'm not entirely sure what this section adds? Certainly spectra weren't mentioned in the methodology section. If there was going to be a discussion of control implications then perhaps these spectra would provide some additional value, but in the current paper version they feel unnecessary.

Line 312 - "Wagner et al. (2010) illustrated how the angle of attack and relative wind speed seen by the airfoil changes with the azimuth position of a rotor blade. For a simple rotor at 8 m s-1 without blade pitch, for veering wind, they found that the angle of attack was larger than for uniform wind while the relative wind speed was lower. The opposite was true for backing." Why is further background information being added into the results section? Surely this should have been discussed in Background and then the analysis foreshadowed in Methodology.

Line 360 – is a 6.5% reduction for the 15 MW turbine to be described as a "significant reduction"?

Conclusions – some threads are pulled together here, but overall the paper feels uncertain as to its specific purpose and aims. I think the whole narrative needs to be clarified and tightened up throughout.

---

## Author Comment (AC1)

**1 Reviewer 1**

**General comments:** The reviewer strongly believes that the paper presents insights into fatigue loading for large floating wind turbines with flexible drivetrains, but not a lot of focus was given to the veer aspect of the study. The results appear original and are well written.

**Page 1: Abstract:** This is a nontechnical review comment – In academic writing, an abstract does not include intext references (e.g Gaertner et al., 2020, Allen et al., 2020, etc). Rewrite to suit.
*Thank you for this comment. The text has been rewritten accordingly.*

**Page 1: Introduction: Line 24:** It would be beneficial to state or quantify the layer in which you are determining the ABL (in the form of an elevation, say, 100-300m or whether it is stable or unstable, or NBL)
*The height of the ABL can vary significantly. Referring to Stull, the text has been updated as: "Another challenge emerges from the rotor spanning a larger portion of the atmospheric boundary layer (ABL). The ABL ranges from 100 m to 3000 m in height, with the smallest heights occurring in stable conditions (Stull, 1988). As the rotor spans more of (or beyond) the ABL, prior assumptions related to the wind profile and wind field are called into question (Veers et al., 2023)."*

**Page 1-3: Introduction:** The introduction so far is well structured. So, the roadmap is clear. However, where you have descriptions like "becomes more uncertain" or "questionable", it would be best to quantify this uncertainty where possible.
*Exact values for these uncertainties are not known, and will of course depend on the turbine, platform, and location of interest. The present work is an attempt to quantify some of these uncertainties. We have added some references. Due to the comments from reviewer 2, the introduction is now significantly altered, and a background section is added.*

**Page 3-10: Introduction:** Make a bold statement about why semisubmersible only was chosen. It will prevent questions about other floating systems.
*The reason for considering semisubmersibles in this study is simply that the availability of a reference floating wind turbine of 22 MW was limited to a semisubmersible at the time when this study was conducted. It was convenient to consider a semisubmersible also for the 15 MW for better comparability. A spar-supported platform, for instance, could be more prone to the yaw-moment induced by wind veer, having less inertia and restoring in yaw DOF. Other floating systems should be studied, and a remark to this effect is added to the conclusions.*

**Page 5: Methodology: Line 106:** The reviewer is interested in how the kxx, kyy, kxy, and kzz is mapped into SubDyn spring elements. Specifically, how you rotate the oA diagonal from the local coordinate frame of the bearing into

SubDyn's element axes. This might interest the general reader too, so consider including it in the paper.

*SubDyn spring elements are specified by means of a cosine matrix. In this work, the (global to local) cosine matrix was obtained by outputting the cosine matrix of the shaft (aligned with the bearings) in SubDyn. The cosine matrix determines the direction of the spring, and spring stiffness entries of each element of the symmetric stiffness matrix were provided in accordance with this cosine matrix. In practice, the spring z-axis (spring element 33) in SubDyn aligns with the shaft axis (x in the coordinate system defined in Figure 2). The spring element y-axis (22) in SubDyn points downwards along local z in Fig 2, and the spring element x-axis (11) in SubDyn points opposite to local y in Fig 2. Off-diagonal terms are matched accordingly considering the local SubDyn and Fig 2-coordinate systems. While we agree that this is useful information, it should be straightforward to apply this when following the SubDyn documentation.*

**Page 3-10: Methodology: Line 115:** Not critical, but is there a reason why the time steps differ between 15MW and 22MW models? Was dt chosen to satisfy stability and accuracy in the Craig–Bampton RO model?

*For the 15 MW, the time step was chosen to satisfy the accuracy of the CB-model (convergence). For the 22 MW, ElastoDyn required a shorter time step than SubDyn to ensure numerical stability, likely because of transients in the beginning of the simulations. We have added this explanation in the revised paper.*

**Page 7: Methodology: Line 147:** Are semisubmersibles not sensitive to directional spreading, since you use a Pierson-Moskowitz spectrum for wave generation. Did you model the wave as unidirectional.

*Long-crested incident waves aligned with the wind propagation direction were considered. This assumption is a simplification, as the focus is here on wind loads. Added some text in the paper to clarify: "The Pierson–Moskowitz spectrum was applied for the generation of long-crested irregular waves aligned with the mean wind direction".*

**Page 9: Methodology: Table 6:** Is there a reason why your grid width (264/316 m) only barely exceeds the rotor diameter?

*Generating turbulent wind fields is computationally demanding. A typical requirement is that the grid should be at least 1.1 times the rotor diameter. The floater does not move much in sway, and therefore the turbulence grid is big enough to cover the rotor during the entire simulation.*

**Page 14-19: Result and discussion:** You present DELs for SV, ST, SVT, and SBT, but without discussing the linearity or complexities of the wind veer cases, readers cannot understand the incremental effect of veer itself. There should be a baseline case (i.e., without veer) for comparison. I believe this is what the study meant by analyzing the influence of veer. This is the comparison the reviewer expects. Otherwise, the topic could simply be "Fatigue loading for

large floating wind turbines with flexible drivetrains."

*Initially, we considered standalone "V", "B", "T" and "S"-conditions. However, as DELs are not linear, DELs from standalone cases cannot be simply superposed, which made the results misleading, and it is not very interesting to consider the influence of veer without turbulence and shear (veer could increase DELs, but this addition would change if turbulence and shear is added). "ST" is therefore here considered the base case, as shear and turbulence are normally accounted for in design (but not veer). A discussion on the effect of veer itself on blade-root loads is given in Sect. 4.2.5, using standalone veer, backing and uniform wind cases, and we have expanded this discussion to look at main bearing and tower top loads. Further, the condition "SV" is removed from the results to focus on the comparison between the backing/veering conditions and the one with turbulence and shear only. Moreover, the title of the paper has been revised and now reads "The influence of wind veer and drivetrain flexibility on fatigue loading for large floating wind turbines" (not only covering veer).*

**Page 18: Result and discussion:** It would be better to keep the y-axis uniform in Figures 12, 13, and 14, as it appears the magnitudes are the same when it is not.

*Thank you for this comment. The figures were not changed in order for the reader to see the importance of different excitation frequencies for a given loadcase, but a sentence was added in the caption to make the reader aware that the y-axis differ between subfigures.*

**Page 20: Conclusion:** Again, to be clear, you rightly discussed the effect of veer, perhaps using your prior knowledge of the topic, but did not provide details on how veer drives DEL increases. The methods and results lack a clear, standalone "veer-only" case and do not show how veer was implemented or isolated in your analysis.

*We have made an attempt to focus more narrowly on veer/backing versus conditions without veer/backing by a) removing the "SV" wind field and b) expanding the discussion on effects of azimuthal variation on load components (including main bearing loads and tower top bending moments).*

**2   Reviewer 2**

Dear authors,

This paper seeks to investigate the influences of drivetrain flexibility and wind veer on wind turbine damage equivalent loads, considered here in the context of blades, tower and main bearings. It undertakes this analysis in the context of large offshore floating wind turbines (22 MW and 15 MW). Results indicate that wind veer can increase DELs by as much as 30-70% for some components under some sets of wind conditions, and that accounting for flexibility of the drivetrain can result in DEL reductions on the order of 6-22% under some conditions. While this manuscript certainly contains interesting results,

I believe extensive revisions are required before it can be deemed ready for publication. My high-level concerns are as follows:

1. The paper lacks a clearly defined and specific set of research goals or questions, along with a clear narrative around why we're interested in this particular set of components, phenomena and metrics. A variety of analyses are undertaken, but the current exposition only links them fairly tenuously. By linking to prior work, motivating what is undertaken, and more narrowly defining objectives the paper can become more focussed, and its contribution made clearer (specific instances of this are highlighted throughout my "specific comments" below). *We have updated the introduction and added a background section discussing prior work and movitation.*

2. Related to the above, various results return only minor sensitivity to the effects under investigation, whereas in some cases the effects are significant. I'd suggest focussing on the latter, and possibly digging deeper in those cases to extract more insight and value.

   *We have removed previous figure 7 and will remove previous figure 10 if the results doesn't change much after running the additional seeds. We have also removed some spectral plots, and expanded some of the discussions.*

3. The paper lacks a thorough treatment of relevant background research and associated contexts. The result of this is a large prior-knowledge burden placed on the reader, and the risk of some readers mis-interpreting findings, results and implication. *A background section is added.*

4. There are various point in the paper where assumptions and modelling decisions need clarification or better explanation/justification.

   *See specific comments.*

5. Finally, I feel some results figures need enhancing (e.g. use all seeds rather than just 1), some seem not to add much value and so might be safely removed (spectra), and some additional plots need adding (mean load plots for MB results). *See specific comments.*

To be clear, I am confident that good work has been undertaken which will be of value to the wind community. But, I believe the current manuscript falls short of what is required from a research publication.

**2.1 Specific comments**

**Abstract** – some bold claims are made here which I'm not sure the results quite live up to. Please consider a slightly more measured summary of paper outcomes in the abstract

  *Thank you for this comment. We have tried to be more specific and measured with regard to the presentation of our results in the revised abstract.*

**General comment** – the title is maybe misleading, as it highlights veer as the headline effect under investigation. Please consider a more representative title for the work which is presented.

*The title is updated to "The influence of wind veer and drivetrain flexibility on fatigue loading for large floating wind turbines"*

**General comment** – there seems to be a general conflation between classic and rolling contact fatigue in the paper (or at least the distinction is never made). These are not the same mechanism!

*In the revised paper, there is now a Background-section, in which we discuss DELs, and we underline that we consider RCF for the main bearings, which differs from fatigue calculated for structural components such as the tower and the blade.*

**General comment** – some amount of drivetrain flexibility is modelled, and its effects on dynamics and loads evaluated. But, it is also important to appreciate that the bearing housing is not modelled. If this component deforms during operation then additional interactions may occur which shorten bearing life. So, when describing what new effects are modelled, I'd encourage you to also highlight those effects which remain un-modelled.

*Thank you for this comment. We have added in the text that some effects are not accounted for, such as the bearing housing.*

**Line 21** – "With increasing turbine size, the applicability of current modelling approaches becomes more uncertain. Mechanisms that have been neglected in the design of smaller turbines may play an important role for larger wind turbines. Increased structural flexibility is one feature that introduces new demands on state-of-the-art analysis tools." Please add some relevant references to back up these claims. This is indeed a much considered topic and so there will be useful citations to include here.

*Thanks for this comment. We added some relevant references here.*

**Line 24** – "Another challenge emerges from the rotor extending deeper into, or even beyond, the atmospheric boundary layer (ABL), so that previous assumptions related to the wind profile and wind field can be questioned." Again, please use some of the many relevant references to back up your claims.

*A reference to Veers et al. was added.*

**Line 25** – "In this work, both of these topics are investigated." I don't think it's valid to claim this analysis specifically considers extension beyond the ABL, and since kinematic (Kaimal spectrum) wind fields are utilised, I'd also advise being careful when making claims about how representative these simulations are of the ABL.

*Thank you for pointing out that the text was unclear. We have removed this sentence and replaced it with an expanded discussion of the work applied.*

**Line 27** – "Based on the assumption of a relatively rigid drivetrain, main bearing response has traditionally been obtained by two separate analyses. First, a global analysis, with the drivetrain represented by a torsional spring and damper, outputs shaft loads. These shaft loads are then combined with either analytical calculations or a detailed local model of the drivetrain to obtain main bearing loads. Analytical calculations are fast but complex to derive for bearings that have off-diagonal or moment-carrying stiffness terms. Local models are accurate but computationally expensive and time-consuming to develop." The text quoted here is, as far as I can tell, the sum-total of background discussion provided on main bearing modelling work. No references are given to back up any of the claims made and, therefore, significant volumes of prior work have been completely ignored. This is highly problematic and leaves the reader without any appreciation for the extant work in this field. An important implication of this is that the reader now also lacks any appreciation for the nuances which remain present in main bearing research, i.e. we don't yet have a clear picture of the dominant failure mechanisms and so much ongoing modelling and data analysis work is specifically in aid of identifying candidate mechanisms (internal to the bearing and/or as a result of inflow conditions and load characteristics). Further work in this domain (e.g. investigating possible veer influences) is only valuable if carefully couched within the full context of main bearing R&D efforts and current understanding.

*We have added background chapters on main bearings and drivetrain modeling, including references.*

Additionally, in the current exposition, it's not clear that the "global analysis" you describe is the aeroelastic-code (FAST, Bladed, Hawc2 etc). Again, clearer writing and proper referencing is needed. For example, you describe the global analysis as representing the drivetrain by a torsional spring and damper; is this true of all/most aeroelastic codes, or is that just FAST?

*We have replaced the phrase "global analysis" with "aero-hydro-servo-elastic analysis" or "aero-elastic analysis". We have also extended the text with a background section including a discussion on drivetrain modeling in various aero-elastic codes.*

Many relevant main bearing papers can be found in WES, Wiley Wind Energy, and Energies journals. The following seem particularly relevant (to be transparent, I am co-author on two of these. Entirely up to you which – if any – you include):

- Electric Power Research Institute (EPRI). (2024) Wind turbine main bearing reliability analysis, operations, and maintenance considerations. Technical report 3002029874.
  https://www.epri.com/research/programs/113055/results/3002029874

- Hart, E., et al. (2023). Main bearing replacement and damage - field data study on 15 gigawatts of wind energy capacity. Technical Report NREL/TP-5000-86228. https://docs.nrel.gov/docs/fy23osti/86228.pdf

- Kenworthy, J., et al. (2024). Wind turbine main bearing rating lives as determined by IEC 61400-1 and ISO 281: a critical review and exploratory case study. Wind Energy, 27(2), 179-197. (also see Zaretsky 2016 referenced therein)

- Sadeghi, F., et al. (2009). A review of rolling contact fatigue, J. Tribol., 131, 041403.

- Tallian, T. E. (1992) Simplified Contact Fatigue Life Prediction Model – Part I: Review of Published Models, J. Tribol., 114, 207–213.

- Feiyu Lu, et al. (2024). A quantitative approach for evaluating fatigue damage under wake effects and yaw control for offshore wind turbines, Sustainable Energy Technologies and Assessments, Volume 66, 103824.

Beyond the lack of exposition concerning prior main bearing literature and modelling work, there is also a lack of similar discussion concerning all components and analysis metrics used. To put this another way, you don't motivate why the study is focussed on blades, towers and the main bearing (when there is extensive literature motivating each as important), and there is no discussion of the nuances of the DEL metric (and for the MB the DEL is based on L10 values, which have another raft of caveats associated with them) and how informative/restrictive it is. These last points all represent significant limitations in our ability to translate simulated loading into expected real world lifetimes and outcomes for components. That doesn't stop the study being valuable, but you have to clearly communicate that these limitations are present. I'd also add that these discussion don't need to be lengthy, and can mostly highlight key points and then refer to relevant literature, but they must be present.

*We have added background sections discussing DELs, main bearings and drivetrain models. In the introduction, we have added some references backing up why we focus on blades and the tower in addition to main bearings.*

**Line 39** – "Wind veer often stems from the influence of frictional forces on the force balance"... what forces balance? "Veer can also be a result of inertial oscillation in the ABL or horizontal temperature gradients" The above two quotes aren't very intelligible to someone not already familiar with these concepts, can you please develop these points so they may be more readily understood.

*In the revision of the introduction, this sentence has been removed.*

**Line 61** – "This paper addresses the influence of increased drivetrain flexibility and wind veer on damage-equivalent loads (DELs) in the tower, blade root and main bearings for two floating turbines" There is no motivation given for why we're interested in drivetrain flexibility and veer in particular? Why these two phenomena in particular?

*We have expanded the introduction to present the previous work by the authors, and discussing why veer and drivetrain flexibility is considered. An ad-*

*ditional background section also provides information regarding motivation and state-of-the-art.*

Building on the above point, I'd add that the introduction is not very well crafted as it currently appears. More specifically, there is a lack of structured narrative to help the reader follow why we're interested in these specific combinations of components and effects, and where we are relative to prior work. One structuring issues example is that the introduction covers drivetrain flexibility and upscaling, then main bearing modelling, then wind veer. We're therefore jumping between big picture and small details, and without ever linking drivetrain flexibility with veer. Please revise the structure and content of the introduction in order to clearly and explicitly arrive at well-motivated research questions that naturally lead into the study which has been undertaken.

*Hopefully the narrative is better now, considering the updates in the introduction, and the additional background section.*

**Line 62** – "A coupled drivetrain-turbine model was built in OpenFAST... etc" This is Methodology or Background content, and should not appear in the Introduction. Indeed, considering this alongside previous comments, perhaps what this paper lacks most is a proper Background section in which past work and relevant literature can be properly presented and discussed.

*We have added a background section, and removed this text from the introduction.*

**Line 85** – "For such large rotors, it is generally recommended to account for large deflections in OpenFAST by modeling the blades using the geometrically exact beam theory in the module BeamDyn. Unfortunately, it was not computationally feasible to combine BeamDyn and the drivetrain model described in Sect. 2.1.2 for the floating turbines". Isn't this a major issue, given the current study is specifically looking to understand loading on large flexible turbines? What if the errors induced by poor blade modelling are swamping any relevant effects from the flexible drivetrain or veer presence? It isn't enough to simply state that it wasn't possible, you need to seek to verify what the impact of this modelling decision will likely be on results. Please consider how this can be done. My suggestion would be to run simulations of both turbine models with the standard/simple drivetrain representation using both BeamDyn and Elastodyn (modal repr.), and then compare blade, tower and main bearing loads coming from each. If those differences are "small", then it provides some confidence that Elastodyn could be sufficient for the elastic drivetrain case. More generally, this simplification and its impact on results needs more thorough treatment in the paper.

*Thank you for this comment. We agree that this is an important drawback. We have followed your suggestion and run simulations with a traditional modal/multibody representation (ElastoDyn) of the tower and shaft comparing results from a model with BeamDyn blades and a model with ElastoDyn blades. We have looked at how this will influence the results related to veer. While the specific numbers differ between the two models, the order of magnitudes are the*

*same, and the results and conclusions related to veer in our paper are representative of the results with BeamDyn blades. We have added a couple of sentences describing this Section 3.1.1.*

**Line 91** – "modified to include a flexible drivetrain, modeled together with the tower" This reads a little like the dynamic multibody interactions would only be between flexible drivetrain and tower. Is there also interaction between the flexible drivetrain and the rotor?

*There is interaction between the flexible drivetrain and rotor too. We added a sentence to explain this.*

**Line 102** – "and a downwind locating SRB" A locating SRB is necessarily double-rowed (e.g. file:///C:/Users/edwar/Downloads/tpi_251_de_en.pdf ). This, and any general discussion of bearings/bearing-types is important contextual information that is currently missing from the paper. Also, why were SRBs chosen, and how were they sized etc... (maybe those details are in a previous paper, but they'd still be relevant here).

*Thank you for this comment. We have added a short discussion on main bearing types for wind turbines in the background section, and the reason for choosing SRBs is presented in the section on drivetrain models (based on discussion with industry, and because the more comprehensive fatigue analyses required for TRBs are outside our scope). With regard to the question on sizing, see the response to your question on the estimation of main bearing stiffness matrices (two questions below).*

**Line 103** – "The diagonals of the main bearing stiffness matrices..." Stiffness matrices have not been introduced, there is therefore a high burden placed on the reader to already know about these concepts.

*We have modified this section to include a description of the stiffness matrices representing the main bearings. We also included some references. Hopefully, this is clearer now.*

**Line 103**– "The diagonals of the main bearing stiffness matrices were estimated using Schaeffler's tool (Schaeffler, 2025) considering the force and deflection of the bearings for a number of different combinations of mean loads" Which bearings? If you have specific bearings being modelled, don't you also have the dynamic capacity and other information which would allow you to calculate L10 lives directly, rather than relying on relative damage? If you can then I believe you should, as then you can also account for the site wind speed distribution and combine all load conditions into a single resultant L10 life. Additionally, that would remove ambiguity between which bearing is expected to fail first (if it fails from rolling contact fatigue).

*We do not have specific main bearings. We have estimated the main bearing stiffnesses based on the estimation of stiffness of smaller main bearings and an extrapolation based on shaft diameter. We therefore do not have L10 lives or dynamic capacities. As discussed below, we have made best guesses on X and*

*Y based on the values of the largest DRSBs available in the product catalogs of some bearing manufacturers. Moreover, there are limitations in applying ISO 281 to bearings sized for such large turbines, and it is better to consider a comparative analysis rather than trying to quantify the fatigue life. We have reframed the text to "To the authors' knowledge, design details of main bearings suitable for turbines the size of 15 MW and 22 MW are not publicly available. In this work, the diagonals of the main bearing stiffness matrices were estimated using Schaeffler's tool considering applied loads and the resulting deflections of several smaller bearings for a number of different combinations of constant loads and then extrapolating to the size of the 15 MW and 22 MW shaft."*

**Table 2** – you utilise stiffness cross-terms, but it's not clear from the explication provided how they are utilised in the flexible drivetrain implementation. Please provide some context.

*We have added a more thorough description of the stiffness matrix representing the main bearing springs, in which the coupling terms/cross-terms/off-diagonal terms are included. Hopefully this gives a better understanding of the main bearing modeling.*

**Line 109** – "Note that the main bearing stiffness estimates differ from those presented by Krathe et al. (2025b) due to updates in estimation methodology". What updates? Are the new values better?

*We discovered a minor error in the estimation methodology for the previous version of the 15 MW and updated the estimates accordingly. Yes, the new values are more appropriate.*

**Line 110** – "Note also that there are large uncertainties in the main bearing stiffness estimates". More fundamentally, bearing stiffness is a function of load. This follows from understanding that under increased load, more rollers are reacting the applied load. This is another point where the bearing context is absent, but would help elucidate the point being discussed. Was the validity of using a single linear spring to model the bearing under different load levels demonstrated in a previous paper? My concern it that the linear assumption could lead to large errors in some load cases. This is why in the past we've coupled our linearised spring model to a bearing contact model which estimates local stiffness as the load changes. For these same reasons, I'm not sure how informative the bearing stiffness sensitivity analysis is here, given the bearing stiffness will in fact change continuously throughout operation. Interpreting that sensitivity analysis should be done with these things in mind.

*The revised text now reads: "Note also that there are large uncertainties in the main bearing stiffness estimates, and that applying constant stiffnesses is in itself an approximation because the main bearing stiffness is a function of load. Therefore, to understand how different main bearing stiffnesses influences the loads, upper, medium and lower stiffness estimates are presented ("high", "medium" and "low"). The majority of the simulations were run with the "medium" stiffness, but in Sect. 4.1.3, the sensitivity of results to main bear-*

*ing stiffness is discussed. This sensitivity test does not give conclusive answers regarding the effects of a load-dependent stiffness, but it provides information regarding the uncertainty due to the applied constant stiffness, and (as will be discussed in Sect. 4.1.3) it indicates how much the main bearings contribute to drivetrain flexibility." We also mention the load-dependent main bearing stiffness in the new background section.*

*It is not possible to apply nonlinear spring stiffness in the linear module Sub-Dyn. Considering the low fidelity of drivetrain models in aero-elastic codes (with the exception of Bladed), accounting for main bearing stiffness is an important first step, and it is reasonable to start with a constant stiffness and increase model fidelity as necessary. Work is ongoing to include the drivetrain flexibility and main bearing stiffness in ElastoDyn and perhaps it will be possible to apply varying main bearing stiffness there.*

**Line 116** – "glue-code (OpenFAST) time step of 0.0015 s and 0.0025 s, respectively." Why these values? I assume for convergence reasons. Did you arrive at these (if so how?) or are they standard values?

*See response to comment from reviwer 1: "For the 15 MW, the time step was chosen to satisfy the accuracy of the CB-model (convergence). For the 22 MW, ElastoDyn required a shorter time step than SubDyn to ensure numerical stability, likely because of transients in the beginning of the simulations." We have added this explanation in the revised paper.*

**Line 134** – If you're interested in veer effects, it could obfuscate the results to also change the power law shear exponent. As it turns out the values across load cases appear to be almost identical, so I don't think this is an issue. But if they were different it would make it difficult to discern what was driving any changes which were observed.

*As the occurrence of veer is highly dependent on atmospheric conditions, it was important for us to obtain representative load cases from the NORA3 data. An analogy could be given for the waves—it would not be realistic to assume similar wave conditions across all mean wind speeds. As you point out, the power law exponent did not differ much between the load cases, and perhaps they could be assumed equal in this case. However, the relevant comparison here is between SVT and ST for the same mean wind speed, rather than comparisons across different wind speeds.*

**Equation 3** – similar to the above comment, if you're interested in veer, don't let TI vary between each load case! (Again, they don't vary much so probably not a huge issue).

*Similarly to the comment above—we wanted load cases that are representative of the environment for this location, and focus on the comparison between cases with and without veer for the same mean wind speed and wave conditions.*

**Section 2.2.3** – shouldn't this be within the previous subsection on Environmental Load Cases?

*Good point. We have combined 2.2.2 and 2.2.3 based on this recommendation.*

**Section 2.3.1** – Please highlight here the fact that DELs allow for comparison between how damaging two sets of conditions are, and that they don't translate to a life estimate for that component. Also important to highlight that the concept for Loads is borrowed from the concept of Damage Equivalent Stress. Importantly, there is no good justification for reformulating the latter as the former, and that this is only truly valid if stress is proportional to the applied load. For turbine blades and tower this is highly likely to be true most of the time. DELs should therefore be highlighted as an indicative tool at best, and one for which we don't currently have a good alternative in general.

*Thank you for the comment. We have added a background section discussing the applicability and drawbacks of DELs.*

**Line 171** – "Neq is the equivalent number of load cycles until failure" Under the calculate DEL

*Thank you for this comment. The sentence has been revised accordingly.*

**Line 173** – "The Wöhler exponent is taken as 10 for the blades and 4 for the tow" reference?

*A reference to DNV-RP-C203 is given for the tower, and a reference to a paper by Jonkman and Matha is given for the blades.*

**Line 181** – "These factors were estimated based on product catalogs" Earlier it seemed like you had known bearings from which you were modelling stiffness. Do you not in fact have that? What is it specifically you are getting from catalogues, I'm guessing the contact angle, or fatigue load limit (e)? Just to be clear, if you look in ISO 281 Table 8 it gives you a formula for e as a function of contact angle.

*As explained in a previous response, we have not considered specific bearing designs. Rather, we have estimated bearing stiffnesses of smaller bearings based on a load-deflection analysis, and extrapolated to the shaft diameters of the IEA 15-MW and 22-MW turbines. Therefore, we do not have e. We refer to the paper by Krathe et al. (2025b), in which the estimation methodology is described: "Detailed bearing design is not the scope of this work, so these quantities were estimated based on bearing catalogues from two manufacturers [26, 27]. Each manufacturer's three largest spherical roller bearings were considered, all with similar fatigue parameters which were also applied here, see Table 6. The reference bearings have an inner diameter of 1200–1700mm, while the IEA 15-MW shaft diameter is 2200mm." The fatigue parameters that we refer to are the radial and axial load factors, meaning that the largest DSRBs of these catalogs had axial and radial factors similar to those represented in Table 7 (in this paper). For clarity, we have revised the text, which now reads "X and Y were estimated based on product catalogs from two bearing manufacturers".*

**Table 7** – For the non-locating bearing Fa = 0 and (assuming contact angle is 0) e=0, so the relevant row should read 1, 0, NA, NA, 0.

*Thank you for this comment. We consider two DSRBs, one locating and non-locating. Axial movements in the non-locating bearing is accommodated by a loose fit between the bearing and its seat (see for instance SKF Bearing Arrangements). We assume that the axial load in this non-locating bearing is zero. The DSRB does not necessarily have a contact angle = 0, and therefore it is more appropriate to say that $F_a = 0$, and use "NA" for $Y$ and $e$.*

**Section 2.3.2** – this treatment of MB DELs lacks any discussion of the L10 life, ISO 281 formulations, or any of the surrounding context, caveats and uncertainties tied in here. Whether or not rolling contact fatigue (surface or subsurface initiated) is a dominant failure mechanism for main bearing in wind turbines remains unknown, beyond that it also remains unclear to what extent ISO lifing methods provide accurate rolling contact fatigue lives given the size and complex nonsteady load conditions of these bearing (especially in large wind turbines). Beyond that, rolling contact fatigue for bearings is expressed as an L10 life, i.e. the time after which 10% of a population are expected to have failed. Another critical factor is the fact that, due to the nature of rolling contact fatigue and the ISO 281 method, main bearing L10 values are driven by the mean MB load rather than load fluctuations (discussed in Kenworthy 2024, listed previously). These are all important and occasionally subtle points which should accompany any application of these methods, to avoid misleading people to believe these metrics can be used to confidently assess main bearing longevity. DEL formulation does allow one to dispense with the bearing dynamic capacity (Cd), but again this should be highlighted to the reader via a proper discussion.

*Thank you for this comment. We have added a discussion on L10 lives and ISO 281 in a background section.*

Finally, equation 7 is simply presented without any derivation or explanation.

*In the text, we refer to the textbook "Shigley's Mechanical Engineering Design", by Budynas and Nisbett. In it, a derivation of the damage equivalent load analogy is provided (referred to there as a "steady equivalent load", "the steady equivalent load that does the same damage as a continuously varying cyclic load". An additional explanation is added in the text.*

Finally if $t_i$ is simply the timestep, then no subscript "i" is necessary.

*In OpenFAST, each time step is equal. However, some other codes, such as SIMPACK, utilizes time steps that changes throughout the simulation. Writing $t_i$ is therefore considered more generic.*

**Section 3.1** – is this a results section? Surely it's more appropriate within the methodology or an appendix?

*Thanks for this comment. The description of the methodology is now moved to a separate section on modal analysis in the Methodology chapter. The results*

*are kept in the results section, because we believe that the influence of drivetrain flexibility on tower and rotor modes is an important finding. We further moved this section into the section "Sensitivity to drivetrain flexibility" to underline that it shows the differences in modal properties as a function of drivetrain flexibility.*

**Line 190** – "Aerodynamics were omitted and mooring lines were replaced by linear springs to facilitate detection of modes. Rigid-body mode natural frequencies were found through decay tests." These statements are made without context, justification or supporting references (if applicable). Again, a high burden is being unfairly placed on the reader here.

*Thanks for the comment. We have added some references here.*

**Figure 4** – Why are we only looking at results for a single seed? Pleased instead develop plots which summarise information across seeds. For example you could easily plot the mean, std, max and min across each set of 6 seeds at each wind speed. This would provide a more robust picture of impacts here.

*We will run the additional simulations and include this information in the plots.*

**Line 218** – "It is difficult to compare main bearing loads between the rigid and flexible..." It feels like we're back into methodology here, why isn't this discussion had in the methodology section?

*We have added a section called "Main bearing loads" in the "Methodology" chapter, and moved this discussion to that section.*

**Line 220** – "As a proxy for main bearing loads, the standard deviations of the shaft axial load and pitch and bending moments are compared among the two models". As mentioned previously, MB rating life (L10) and hence $DEL_{MB}$ is principally driven by the mean MB load and not by standard deviations. This is counterintuitive, and why a detailed treatment of bearing and rolling contact fatigue complexities are essential. Please undertake the same analysis (summarising across all seeds) for the mean loads as well as std.

*We will run the additional simulations to account for all seeds and include mean values in the plot. For the one seed currently investigated, shaft mean loads does not change much with drivetrain flexibility (max 1.5% for the yaw moment, and max 2.4% for the pitch moment (but the pitch moment difference is for LC2, for which the tower natural frequency is shifted so that we don't consider this result that valuable)). While this study is about DELs based on main bearing rating lives, as already pointed out by reviewer 2, there is ongoing investigation to understand whether ISO 281 rating lives can predict main bearing failure. Because of this uncertainty, we also believe it is valuable to understand how the dynamic main bearing loads change, and therefore also present main bearing load standard deviations. We have made an attempt to describe this issue in a section on main bearing loads in the methodology chapter and argue why we also want to understand how main bearing load standard deviations are influenced*

*(not only mean).*

**Figures 4 and 5** – When revising these figures, please use the same fixed axes on all plots in order to make them directly comparable to each other without careful study of axis scaling.

*This change has been implemented in the revised text.*

**Section 3.2.2** – This section is more of a verification that the uncertainty around bearing stiffness is likely not driving large errors in the results. As such I'd suggest moving it to an appendix, as it distracts from the main focusses of the paper.

*Thank you for the comment. We believe that this section provides useful information with regard to how much main bearing flexibility contributes to the changes in DELs and loads compared to the flexibility of the rest of the drivetrain. Responses do not change much with increased main bearing stiffness. This indicates that structural flexibility of the bedplate and shaft contribute more to the differences in response between a rigid and flexible drivetrain than main bearing stiffness. We have added some text discussing this.*

*Although we do not account for varying main bearing stiffness, it is interesting to understand how the main bearing and global response change with the main bearing stiffness—because there is uncertainty in the estimation of the constant (and varying, if such was applied) stiffness. To a certain extent, this section also gives an idea of how the varying stiffness would influence the DELs (if the loads showed no sensitivity to bearing stiffness at all, is it reasonable to think that modeling varying stiffness is important?).*

**Figure 7** – as mentioned above, it is mean loads that are mostly going to drive your MB DEL. Please also develop figures showing changes in mean loading as you change bearing stiffness. This should include Fa, Fr and P values. As above, it seems this has only been plotted for a single wind seed at each point, please therefore also represent results for all seeds here as well.

*We will run the additional simulations to account for all seeds. For the one seed considered, the mean axial loads were not influenced by drivetrain flexibility, while radial loads were influenced by less than 0.5% for the downwind bearing and less than 0.1% for the upwind bearing, making graphical representations of these results rather uninteresting. The sentence has for now been updated to: "Main bearing DELs and mean loads (not plotted) are even less affected, with a maximum difference of 0.5 %.". The text will be revised if additional seeds alter these quantities.*

**Line 254** – what is the logic behind including a case for shear and veer without turbulence? I can't see why that adds anything?

*Thank you for this comment. The "SV" wind field was useful to say something about how important turbulence is for the DELs. However, we agree that this investigation is out of scope for this paper and have removed these results.*

**Section 3.3.4** – I'm not entirely sure what this section adds? Certainly spectra weren't mentioned in the methodology section. If there was going to be a discussion of control implications then perhaps these spectra would provide some additional value, but in the current paper version they feel unnecessary.

*We respectfully disagree. Spectral analysis is a common tool applied to evaluate responses of aero-elastic simulations. It allows us to understand the sources of differences between load cases and wind fields, and to compare different sources of excitation to each other (turbulence, waves, 3P, natural frequencies etc.). We have however, removed the appendix containing additional plots, and parts of the blade results, and we have expanded the discussion to make sure it adds value to the paper. Additionally, a section briefly introducing the methodology of spectral analysis is added.*

**Line 312** – "Wagner et al. (2010) illustrated how the angle of attack and relative wind speed seen by the airfoil changes with the azimuth position of a rotor blade. For a simple rotor at 8 m/s without blade pitch, for veering wind, they found that the angle of attack was larger than for uniform wind while the relative wind speed was lower. The opposite was true for backing." Why is further background information being added into the results section? Surely this should have been discussed in Background and then the analysis foreshadowed in Methodology.

*This text is split between the introduction and the Methodology section, where a new subsection is added.*

**Line 360** – is a 6.5% reduction for the 15 MW turbine to be described as a "significant reduction"?

*We have rephrased this sentence to "Including the flexible drivetrain in the global analysis leads to significant reductions in tower-top torsional and fore-aft damage-equivalent bending moment for the 22 MW turbine (22 %). Reductions are also seen for the 15 MW turbine (6.5 %).".*

**Conclusions** – some threads are pulled together here, but overall the paper feels uncertain as to its specific purpose and aims. I think the whole narrative needs to be clarified and tightened up throughout.

*We have made some updates to the conclusion, specifically on main bearing DELs.*